# Machine Failure Prediction Using Survival Analysis

**Dimitris Papathanasiou** [1,*]🆔 **, Konstantinos Demertzis** [2,*]🆔 **and Nikos Tziritas** [3]

1. Department of Computer Science and Biomedical Informatics, University of Thessaly, Papasiopoulou 2–4, 35131 Lamia, Greece
2. School of Science and Technology, Informatics Studies, Hellenic Open University, Aristotle 18, 26335 Patra, Greece
3. Department of Computer Science and Telecommunications, University of Thessaly, 35100 Lamia, Greece
* Correspondence: padimitrios@uth.gr (D.P.); kdemertz@fmenr.duth.gr (K.D.)

**Abstract:** With the rapid growth of cloud computing and the creation of large-scale systems such as IoT environments, the failure of machines/devices and, by extension, the systems that rely on them is a major risk to their performance, usability, and the security systems that support them. The need to predict such anomalies in combination with the creation of fault-tolerant systems to manage them is a key factor for the development of safer and more stable systems. In this work, a model consisting of survival analysis, feature analysis/selection, and machine learning was created, in order to predict machine failure. The approach is based on the random survival forest model and an architecture that aims to filter the features that are of major importance to the cause of machine failure. The objectives of this paper are to (1) Create an efficient feature filtering mechanism, by combining different methods of feature importance ranking, that can remove the "noise" from the data and leave only the relevant information. The filtering mechanism uses the RadViz, COX, Rank2D, random survival forest feature ranking, and recursive feature elimination, with each of the methods used to achieve a different understanding of the data. (2) Predict the machine failure with a high degree of accuracy using the RSF model, which is trained with optimal features. The proposed method yields superior performance compared to other similar models, with an impressive C-index accuracy rate of approximately 97%. The consistency of the model's predictions makes it viable in large-scale systems, where it can be used to improve the performance and security of these systems while also lowering their overall cost and longevity.

**Keywords:** machine failure; survival analysis; random survival forest; feature analysis; feature selection

## 1. Introduction

Critical infrastructure systems, such as water supply, power supply, transportation, telecommunications, etc., play a significant role in the sustainable development of modern societies. Modern infrastructure systems are highly interconnected and consist of geographically extensive networks. Continuous communication and data exchange between these systems leads to interdependencies that are essential for their proper functioning and the functioning of the overall system they belong to. Due to the large-scale networking of infrastructure systems, there can be economic, social, health, and environmental problems in case of their failure. The failure of these systems can arise from extreme natural phenomena (hurricanes, floods) or technological disasters and cyber-attacks. As a result, systems of this type must be regularly monitored, upgraded, and maintained [1].

Ensuring the healthy and continuous operation of systems such as aircraft engines, cars, computer servers, and even satellites, is an imperative need, given their contribution to critical services, beyond urban infrastructure. The accurate prediction of their malfunctions and, by extension, their operational interruptions, can contribute to improvements in the design of proactive fault-tolerant systems, as well as significant cost reduction through

prompt fault reporting. Previous research has used various techniques to create predictions in scenarios such as the autoregressive model [2], principal component analysis [3], and opposite degree algorithm [4].

Furthermore, a significant amount of research has been carried out in the anomaly detection field, which has a closely correlated relation to machine failure. The research involves a variety of proposed models mostly making use of different machine learning techniques such as ANN [5,6], RF and SVM [7], and convolutional neural networks (CNNs) and long short-term memory (LSTM) [8].

To accurately predict machine failure and prevent costly downtime, it is essential to utilize a reliable and flexible approach that can account for a wide range of factors influencing machine degradation. While traditional statistical models have been widely used in failure prediction [9], they often rely on strict assumptions about machine degradation patterns that may not accurately reflect the real-world complexity of the problem. In contrast, survival analysis has emerged as a promising approach, offering several advantages that can help improve the accuracy and efficiency of predictions. By incorporating time-to-failure information, handling right-censored data, accounting for covariate effects, and providing flexibility in application, survival analysis represents a superior alternative to traditional models in the context of machine failure prediction.

This research aims to address a gap in the literature by exploring the use of survival analysis in combination with various feature selection/analysis and machine learning methods to predict machine failure. While some studies have utilized survival analysis to predict failure, few have examined the effectiveness of different feature selection/analysis methods in conjunction with this technique. Most of the existing research focuses primarily on different machine learning models, such as LSTM networks [10,11], moving away from the survival analysis approach. While machine learning models are undoubtedly useful in predicting machine failure, survival analysis may be more appropriate when the goal is to predict failure times and identify key factors in failure prediction. In this proposed model, a machine learning survival analysis technique is used, along with feature analysis and selection methods. The machine learning model used is random survival forest (RSF), which is well-suited to time-to-event data, such as machine failure times, and has several advantages over other machine learning methods in the context of survival analysis. Examples include the ability to handle time-dependent covariates, non-linear relationships between covariates and survival, and interactions between covariates [12].

Unlike survival analysis models, standard machine learning algorithms are not equipped to handle the right-censored data that are prevalent in machine failure datasets. RSF is a great way to incorporate machine learning and simultaneously overcome this problem, due to its ability to handle high-dimensional, complex data [13]. RSF can handle both continuous and categorical predictors, as well as complex interactions, nonlinear relationships, and time-varying effects. This makes it particularly useful in survival analysis settings where there may be many potential predictors that interact in complex ways [14]. One of the key benefits of using RSF is its ability to handle missing data, which is common in many real-world datasets [13–15]. RSF uses a tree-based approach to impute missing values by splitting the data at each node based on the available data, and then using the available data to make a prediction for the missing value [13]. This imputation process is repeated multiple times, resulting in a distribution of imputed datasets that can be used to estimate uncertainty. Overall, the flexibility and versatility of RSF make it a powerful tool for survival analysis, particularly in situations where there are many potential predictors and complex interactions among variables [13].

Our research makes the following contributions:

- We propose a new machine failure prediction model aimed at increasing prediction accuracy using machine learning mechanisms and the method of survival analysis;
- The proposed model includes a data filtering/selection layer that is designed to choose the most suitable features for training the machine learning model. This is achieved by

implementing various feature analysis/selection techniques. Each of the techniques employed targets a distinct aspect of comprehending the data;

- A random survival forest model is used to combine machine learning and statistical analysis, overcoming their limitations, that affect the prediction [12];
- We objectively examine the model on a collection of 10,000 machine instances, with experiments showing promising statistical accuracy;
- Our work contributes to the existing literature on the use of RSF for machine failure prediction, as well as the need for a strong feature extraction level for model training.

The innovation presented in this paper centers around a novel approach to predict machine failure in large-scale systems, by incorporating a technique that prioritizes the suitability of the data. The proposed method utilizes the RSF model and an efficient feature filtering mechanism, incorporating a range of feature importance ranking techniques to eliminate irrelevant data and retain only pertinent information. This process effectively eliminates "noise" from the data, ultimately leading to increased accuracy and consistency in the prediction outcomes.

The overall organization of the paper is as follows. Section 2 provides an overview of the related work that was reviewed for this research, including a comparison of our proposed approach to other proposals in the literature. In Section 3, we present the mathematical and theoretical background of the techniques utilized in this study. In Section 4, we describe the model created in our study as well as the dataset used for training and testing the model. Additionally, we discuss the different failure modes of the machines in the dataset. In Section 5, we present the results and findings of our study, which are later discussed in Section 6. Finally, Section 7 concludes and closes the paper.

## 2. Related Work

Despite the abundance of statistical methods that can be used, survival analysis is conceptually largely aligned with the study of predicting the failure of a machine [16].

Kaplan–Meier curves and the COX regression model have been employed in similar research to determine the relationship between the survival time of a subject and one or more prognostic variables. In [17], the SMED (single-minute exchange of die) philosophy and survival analysis are used to reduce transition times. In this research, the COX model is also used to identify the significance of causes of time loss. The proposed methodology predicts activity times, considering only the characteristics that were identified as significant toward transition times, which is defined as a limitation by the authors.

Similarly, the model proposed in [18] follows a gradual Bayesian approach to model failure using the tree-like accident theory and the Bayesian survival analysis model to predict the probability of survival for welded pipes. Using Bayesian, Kaplan–Meier, and Weibull curves, the authors construct staged Bayesian distribution, which is then used to make predictions about the time-to-failure of the pipes. Weibull distribution is also used in [19] to predict the life of battery cells, combined with the exponential, log-normal, and log-logistic distributions to create an accelerated failure time (AFT) parametric survival model. In this research, the authors concluded that low values of prediction error could be achieved by only using a small number of variables on the proposed model. The authors assert that this discovery holds significant value in their research, as their model yielded a total decrease of 40% in the root mean square error (RMSE). A limitation of this study is that the authors relied on only two datasets to support their findings, without exploring the use of other datasets.

In [10], an LSTM approach is presented for the remaining life prediction of machines. The proposed approach leverages the advantages of LSTMs in capturing temporal dependencies in sensor data while also effectively handling missing data. The authors conducted experiments on a real-world dataset of a milling machine and evaluated the performance of their novel approach in comparison to various baseline methods. The results indicate that the LSTM-based approach surpasses the other meth-

ods in accurately predicting the machine's health status and effectively capturing its dynamic behavior. Out of the variety of models tested, the bidirectional-LSTM model had a total RMSE value of 15.42 cycles, outperforming all other models, such as deep convolutional neural network (DCNN) (18.44 cycles); support vector regressor (SVR) (20.96 cycles); multilayer perceptron (MLP) (20.84 cycles); bidirectional recurrent neural network (BD-RNN) (20.04 cycles); and a classic LSTM (18.07 cycles). Similarly, the authors of [11] proposed a semi-supervised deep architecture for predicting the remaining useful life (RUL) of turbofan engines. The model uses both labeled and unlabeled data to enhance its performance and reduce the need for extensive labeled data. It involves a combination of a convolutional neural network (CNN) and a LSTM network that work in tandem to extract features and capture the temporal dependencies of the input data. The model was evaluated on the C-MAPSS dataset and compared against several state-of-the-art methods, achieving superior performance in terms of both RUL prediction accuracy and mean absolute error. Specifically, the model proposed by the authors yielded superior RMSE results for most of the subsets that it tested, with the value of 12.10 on the FD003 subset being the lowest, while also providing the best prediction result on all of them (FD001: 231, FD002: 3366, FD003: 251, FD004: 2840). A limitation of the study, as stated by the authors, is the use of a piece-wise linear degradation model, which does not account for the individual degradation patterns of each engine in each subset. The authors plan to address this limitation in future work by exploring the use of an unsupervised fault detector based on a variational autoencoder to optimize performance.

Introduced in [20] is a method based on DCNNs to diagnose faults in induction motors using multiple signals. The proposed method leverages the advantages of DCNNs in automatic feature extraction and achieves improved diagnostic performance by combining information from multiple sensor signals. The authors conducted experiments on a dataset containing multiple types of faults in induction motors and evaluated the performance between two different architectures of their proposed method. The first architecture utilized a multichannel model that merged two separate time–frequency images from vibration signals and current signals, forming a two-channel image. This image was then fed into a deep model that consisted of three 2D convolutional layers and a fully connected layer with ReLU activation functions. The output layer had six units that correlated with six distinct labels. The second architecture used two convolutional networks were utilized to analyze different sensor signals separately, and then merged in fully connected layers to contribute to the output of label prediction. One network was trained on vibration signals, while the other was trained on current signals. The learned fault signatures from each network were combined by flattening them into a fully connected layer with 1024 ReLUs. The output layer used for predicting the state label was the same as the one used in architecture 1. The confidence interval analysis showed that the proposed multi-signal DCNN model had stable performance and the merged model outperformed the multi-channel model, with a 95% likelihood of covering fault classification skill between 99.89% and 99.93%. To address the issue of limited training data for deep architectures, the authors suggest the use of data augmentation techniques to expand the dataset and exploring pre-existing models for fault diagnosis as fields of improvement for their future work.

The paper [21] proposes a method for equipment failure diagnosis that addresses the challenge of limited data and imbalanced data distribution. Specifically, the proposed method combines the synthetic minority oversampling technique (SMOTE) with a conditional tabular generative adversarial network (CTGAN) to predict equipment failures with a mixture of numerical and categorical data. The experimental results show that the proposed method outperforms other similar methods in five-category failure classification, even when failure data account for less than 1% of the total data. The proposed model showed a high recall rate of 0.9068, an accuracy of 0.8712, and a balanced accuracy of 0.8883. The recall rate and balanced accuracy were the highest across all methods tested by the

authors which, apart from the crated model, were a CatBoost (non-oversampling) model, a combination of SmoteNC and CatBoost, and finally, another combination of the ctGAN and CatBoost models. It is noteworthy that the highest accuracy was obtained using the CatBoost algorithm without oversampling. Moreover, the paper highlights the importance of false positives in equipment failure prediction, as the cost of sudden machine downtime far exceeds that of system misdiagnosis. Therefore, the proposed method aims to increase the possibility of false positives to reduce the possibility of false negatives. The authors also note that the interpretability of the equipment failure prediction results is crucial, and they incorporated a tree-based model for failure prediction to analyze the causes of failures and implement preventive measures accordingly.

In [22], the authors make a comparative study to evaluate a plethora of machine learning techniques for the task of fault detection and classification. The models used in this study are SVM classifier, KNN classifier, random forest, logistic regression, and decision tree. All models were tested on five datasets, and their accuracy and AUC-ROC scores were measured. The authors concluded that the best performing machine learning method was random forest, with an average accuracy of 0.964 and an average AUC_ROC score of 0.948 across all datasets. The other notable methods were the decision tree model, with an average accuracy of 0.959 and AUC_ROC score of 0.944, and KNC, with an average accuracy and AUC_ROC score of 0.942 and 0.930, respectively.

In the proposal of [12], the authors suggest a new approach for predicting the remaining service life of water mains by combining machine learning and survival statistics. The authors developed a machine learning algorithm that uses a combination of historical failure data and pipe-specific characteristics to predict the probability of failure at any given time. They then applied survival statistics to estimate the remaining service life of the water main based on the predicted failure probability. The study utilized two distinct machine learning models—specifically, a random forest model and a random survival forest model—and additionally incorporated the Weibull proportional hazard survival model to assess and compare their respective abilities, in order to accurately predict the remaining useful life of water mains. The results showed that the RSF model achieved superior performance (C-index = 0.880) compared to the Weibull proportional hazard survival model (C-index = 0.734) and the random forest machine learning model (C-index = 0.807), indicating the potential of machine learning in predicting the remaining service life of water mains.

The literature reviewed in this study indicates that many of the methods for predicting remaining useful life either solely employ survival analysis [16,18,23] or only use machine learning techniques [10,11,20]. However, combining both approaches can be beneficial, as demonstrated by the papers [12,19], which use a combination of survival analysis and machine learning to make their predictions. Furthermore, most of the papers using survival analysis rely on the use of the COX model for their feature evaluation [18,19,23,24]. By relying solely on the COX model for feature selection, important non-linear or time-dependent relationships between predictor variables and survival time may be missed or obscured, leading to a potentially incomplete or inaccurate understanding of the underlying data. Similarly, in the case of [12], only using feature ranking from a RSF model may not provide information about the direction or magnitude of the relationship between predictor variables and survival time. That is where the combination of different feature ranking/selection techniques can prove to be an advantage in our model. By not only using the standard COX and RSF feature ranking/selection methods, we can achieve a more comprehensive and accurate understanding of the data, as well as increased confidence and validation of the results, while mitigating some of the limitations and biases of each individual method.

## 3. Background

### 3.1. Survival Analysis

#### 3.1.1. Kaplan–Meier and Nelson–Aalen Methods

The Kaplan–Meier estimator is a non-parametric statistical method used to calculate the survival function. The survival function indicates the probability that the subject participating in the study will survive beyond a certain period [12]. In the case of this study, the event is the time until the machine ceases to function. The probability of survival for each time point is calculated using the following formula:

$$S = \frac{NSL - NSD}{NSL} \tag{1}$$

where $NSL$ is the number of machines that function at the start and $NSD$ is the number of machines that stop functioning.

The Nelson–Aalen estimator is used for the same purpose as the Kaplan–Meier estimator, to summarize and display data. Unlike Kaplan–Meier, the Nelson–Aalen estimator uses the hazard function instead of survival [12]. The hazard function focuses on the rate of occurrence of the event we want to observe at time t and is defined by the formula:

$$\widehat{H(t)} = \sum_{t_i \leq t} \frac{d_i}{n_i} \tag{2}$$

where $d_i$ is the number of events at time $t_i$ and $n_i$ is the number of machines at time $t_i$.

Both the Kaplan–Meier and Nelson–Aalen estimators are commonly used in survival analysis to estimate the survival function and hazard function, respectively. These methods can provide valuable insights into the time-to-event data and can be used to compare survival between different groups or treatments.

#### 3.1.2. COX Proportional-Hazards Model

The COX proportional hazards model is a regression model commonly used by researchers to determine the relationship between a subject's survival time and one or more predictor variables. It provides insights into how various parameters affect the duration of a subject's survival. Unlike the metrics mentioned previously, the COX model allows for calculations to be made considering more than one variable. Furthermore, the model can be used for both categorical and non-categorical variables, in contrast to Kaplan–Meier and Nelson–Aalen, which can only be used for categorical variables. The COX model can be used to identify how different factors in the dataset affect the event of interest. The hazard function used to calculate COX is:

$$h(t) = h_0 \cdot e^{(b_1 \cdot x_1 + b_2 \cdot x_2 + \cdots + b_n \cdot x_n)} \tag{3}$$

where $t$ is the survival time, $h(t)$ is the hazard function, $(x_1, x_2, \ldots, x_n)$ are the variables, and $(b_1, b_2, \ldots, b_n)$ are the regression coefficients of the variables. The values $e^{b_i}$ are known as hazard ratios (HR), and they are used to measure the influence of the variables. HR = 1 $\Rightarrow$ no effect, HR $\preceq$ 1 $\Rightarrow$ decrease in risk, HR $\succ$ 1 $\Rightarrow$ increase in risk.

The COX proportional hazards model has several advantages over other survival analysis methods. One advantage is that it can handle both continuous and categorical variables simultaneously, allowing for the analysis of multiple variables. Furthermore, the COX model does not make any assumptions about the distribution of survival times, making it more robust to violations of assumptions. Additionally, the model provides information on the direction and magnitude of the effect of each predictor variable on the hazard of the event.

#### 3.1.3. Survival Trees

Survival trees (ST) are decision trees that are specifically designed for analyzing time-to-event data. In the context of survival analysis, the goal of building a survival tree is to

identify subgroups of instances which differ in their risk of experiencing the event, based on their baseline characteristics [25].

The basic idea of a survival tree is to recursively partition the sample into increasingly homogeneous subgroups with respect to the event of interest. The partitioning is achieved by repeatedly splitting the data based on the values of one or more predictor variables, such that the within-node heterogeneity in terms of the survival outcomes is minimized [25]. At each step of partitioning, the algorithm selects the variable and the split point that maximally differentiates the survival outcomes of the subgroups defined by the split. This process continues until no further improvement in within-node homogeneity is achieved or a stopping criterion is met. The result is a tree structure where each terminal node represents a distinct subgroup with a unique risk profile for the event of interest.

Once the tree is built, it needs to be pruned to prevent overfitting and improve its generalizability to new data. The most used method for pruning is cost complexity pruning, which involves adding a penalty term to the impurity measure to favor simpler trees that avoid overfitting. Other methods, such as cross-validation or resampling techniques, can also be used for pruning.

Survival trees have several advantages over traditional regression models, including their ability to handle nonlinear and interactive effects, identify distinct subgroups with different risk profiles, and provide easily interpretable results in the form of a decision tree. However, they also have some limitations, such as their tendency to create overfit trees, sensitivity to the choice of split criterion and stopping rule, and inability to handle time-varying covariates or competing risks.

### 3.1.4. Random Survival Forest

RSF is a machine learning method for predicting survival outcomes, extending the classical random forest (RF) algorithm to handle censored survival data. RSF constructs an ensemble of decision trees, where each tree is grown using a random subset of the data and a random subset of the features. To handle censored data, RSF introduces a new splitting criterion that considers the distribution of the survival times in each node of the tree. Specifically, the splitting criterion is based on the log-rank statistic, which measures the difference in survival times between two groups of observations [13].

The log-rank statistic is used to measure the difference in survival times between two groups of observations. It is calculated as:

$$LR = \frac{(O - E)^2}{V} \tag{4}$$

where $O$ is the observed number of events in the group, $E$ is the expected number of events based on the Kaplan–Meier estimator, and $V$ is the variance of the number of events based on Greenwood's formula.

The splitting criterion in RSF is based on the log-rank statistic. At each node of the tree, the algorithm considers all possible splits on all possible features and selects the split that maximizes the log-rank statistic. Specifically, the splitting criterion is defined as:

$$S = \frac{\left( LR_{left} - LR_{right} \right)}{SE} \tag{5}$$

where $LR_{left}$ and $LR_{right}$ are the log-rank statistics for the left and right child nodes, and $SE$ is the standard error of the log-rank statistic.

Once the tree ensemble is constructed, RSF can make predictions for new observations. The predicted survival probability for an observation is calculated as the average of the

survival probabilities predicted by all the trees in the ensemble. The survival probability for a given time t is estimated as:

$$P(t|x) = e^{-H(t|x)} \tag{6}$$

where $H(t|x)$ is the hazard function predicted by the tree ensemble for the observation with covariate x.

### 3.2. Feature Analysis

Feature analysis is a process of examining the input features in a prediction model to gain insights into their predictive power and their relationship with the target variable. It is an important step in the development of prediction models, as it can help to identify the most important features for predicting the target variable and to understand the underlying patterns and relationships in the data. By analyzing the feature importance scores and/or combining them with other techniques, one can gain insights into the factors that are most strongly associated with the, in our case, time-to-event outcome and develop more accurate prediction models.

#### 3.2.1. RadViz

The RadViz method is a visualization technique that is used to represent the relationship between multidimensional data points in a two-dimensional space. It maps each data point onto a circle with a set radius and then draws lines from the center of the circle to each point. The position of each data point on the circle is determined by a weighted average of its values in each dimension. The formula for calculating the position of a data point on the circle is as follows:

$$x_i = \frac{\sum_j (w_j \cdot v_{ij})}{\sum_j w_j} \tag{7}$$

where $x_i$ is the position of the data point on the circle, $v_{ij}$ is the value of the data point in the $j_{th}$ dimension, and $w_j$ is a weight that determines the importance of the $j_{th}$ dimension. The weights are determined by the user and are used to emphasize certain dimensions over others. Once the positions of the data points on the circle are determined, lines are drawn from the center of the circle to each point. The length of each line is proportional to the distance between the data point and the center of the circle, which represents the overall relationship between the data points [24].

The RadViz method provides an intuitive visualization of how different dimensions contribute to the overall relationship between the data points, making it a useful tool for exploratory data analysis and for communicating complex relationships between data points.

#### 3.2.2. Rank2D

The Rank2D model is a feature selection technique that ranks input features based on their pairwise correlation with the target variable. The model used in our study utilizes two types of ranking methods: Pearson ranking and covariance ranking. Pearson's ranking is based on the Pearson correlation coefficient, which measures the linear relationship between two variables. Covariance ranking, on the other hand, measures the linear relationship between two variables without standardizing their scale. The Rank2D model first computes the pairwise Pearson correlation coefficients and covariance coefficients between each input feature and the target variable. The features are then ranked based on their absolute value of the correlation coefficient or covariance coefficient. The top-ranked features are then selected for use in the classification model. The mathematical formula for Pearson ranking is given by:

$$r(x,y) = \frac{n \cdot \sum (x \cdot y) - \sum x \cdot \sum y}{\sqrt{(n \cdot \sum x^2 - (\sum x)^2)(n \cdot \sum y^2 - (\sum y)^2)}} \tag{8}$$

where $r(x,y)$ is the Pearson correlation coefficient between input feature $x$ and target variable $y$, $\Sigma$ is the sum of the values of the input feature or target variable, and $n$ is the total number of data points. The mathematical formula for covariance ranking is given:

$$cov(x,y) = \frac{\sum_i (x_i - mean(x)) \cdot (y_i - mean(y))}{n-1} \tag{9}$$

where $cov(x,y)$ is the covariance between input feature $x$, and target variables $y$, $x_i$, and $y_i$ are the values of input feature $x$ and target variable $y$ for the $i_{th}$ data point; $mean(x)$ and $mean(y)$ are the mean values of input feature $x$ and target variable $y$; and $n$ is the total number of data points.

Rank2D can be used to select the most relevant features for classification tasks and can improve the accuracy and efficiency of the classification model. The model can be particularly useful in situations where the number of input features is large and feature selection is required to reduce the complexity of the model.

### 3.3. Feature Selection

Feature selection is a critical step in the development of machine learning models, such as random survival forest. It involves identifying a subset of relevant features from a larger set of input features. Optimal feature selection can lead to significant improvements in the performance of classification models, as well as providing insights into the underlying patterns and relationships in the data. Feature selection is crucial for several reasons, such as reducing the dimensionality of the problem, improving the interpretability and generalization performance of the model, and reducing the computational cost of training and prediction [26]. Therefore, proper feature selection techniques are essential to develop effective machine learning models that can accurately solve real-world problems.

Recursive Feature Elimination

Recursive feature elimination (RFE) is a popular feature selection technique that recursively removes less important features from a dataset until a desired number of features is reached. RFE uses a machine learning algorithm to rank the importance of each feature and iteratively eliminates the least important features until the desired number of features is selected. The basis for RFE involves training a machine learning model on a subset of the features and evaluating the importance of each feature based on its contribution to the model's performance. The RFE model can improve the accuracy and efficiency of a prediction model by identifying the most important features, while also reducing the risk of overfitting [26].

The results of RFE can be important in feature selection, as they can provide valuable insights into the underlying patterns and relationships in the data. By identifying the most important features, RFE can help to simplify the prediction model and improve its interpretability, while also reducing the computational complexity of the model.

## 4. Materials and Methods

### 4.1. The Proposed Model

The proposed prediction model is based on survival analysis for processing and extracting conclusions in a human decentralized context, focused on non-biological entities. Survival analysis concerns predicting the time until an event occurs. In this specific study, our interest is focused on using failure data to estimate the machine's life span under specific usage and stress conditions. The strategy followed for creating the model is:

1. Examination of the significance of the functional characteristics of the machines;
2. Selection of the characteristics that are of statistical and practical importance for the prediction model;
3. Feeding these data into the random survival forest model for training and the ability to predict with the best possible accuracy.

Model Workflow Description

As presented in Figure 1, the initial step is to extract the helpful features from the dataset, using the RadViz, Rank2D, COX, and feature ranking of RSF. Each of the methods is used to achieve a different part of understanding of the data. RadViz visualization technique is used for the identification of clusters of highly correlated features with the survival of the machine. Rank2D, on the other hand, enables the identification of correlations among the features. The COX model is a suitable method for determining the features that exert the greatest impact on the survival time of the machine. Meanwhile, the feature ranking of RSF can model non-linear relationships between the predictors and the outcome, making it a valuable tool for feature selection in this domain.

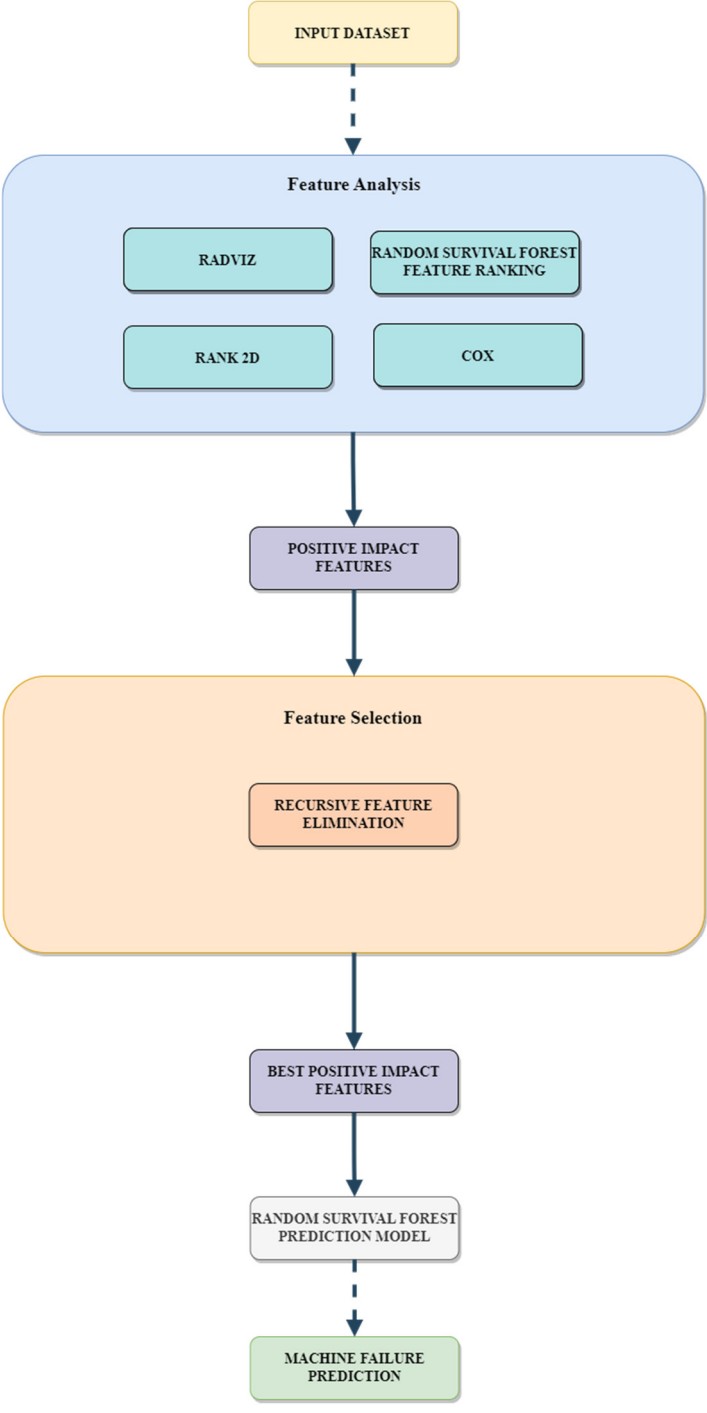

**Figure 1.** Model workflow diagram.

Following the feature analysis, an RFE model is employed to identify the optimal set of features that have a significant impact on machine failure prediction. This step is crucial to ensure that the RSF model is trained with the most relevant and informative features, leading to improved predictive performance. By selecting the best subset of features using the RFE approach, the RSF model can effectively capture the complex relationships between the predictors and the survival outcome, leading to accurate and reliable predictions.

The features that are deemed important through the various layers of feature analysis and selection are utilized in the training of the RSF model. By incorporating only the top-performing features, the RSF model can achieve a higher level of accuracy and reliability in predicting machine failure. The trained model can then be used to make predictions on new datasets and aid in proactive maintenance, ultimately reducing downtime and costs associated with machine failure.

### 4.2. Dataset Description

The dataset used in this study [27] was provided by Stephan Matzka from the School of Engineering–Technology and Life at Hochschule für Technik und Wirtschaft Berlin, Germany. The dataset comprises 10,000 rows, each containing 14 columns of features, including a unique identifier, product ID indicating the quality variant, and variant-specific serial number. Additional features include air temperature, process temperature, rotational speed, torque, tool wear, and a machine failure label. The product ID denotes the quality variant as either low, medium, or high. The air temperature and process temperature in a manufacturing process are generated using separate random walk processes with different standard deviations. The rotational speed is calculated from a power value with added noise, and torque values follow a normal distribution with no negative values. The quality variants add different amounts of tool wear to the process. The machine failure label indicates whether the machine has failed due to one of five failure modes, each with specific conditions, but it is not disclosed to the machine learning algorithm which failure mode has caused the failure.

The right-censored items in the dataset, meaning the machines that did not fail during its composure, are presented in Table 1. The percentage of the censored machines was a high 96.61%, which declares an imbalance in the dataset. The high proportion of right-censored instances in a survival dataset can have a significant impact on the performance of prediction mechanisms such as RSF, due to the potential lack of information about the failure times of non-censored instances. As the RSF model is designed to handle right-censored data, it can still provide reasonable estimates of the survival function and hazard rate. However, the imbalance in the dataset can result in a biased model that is more accurate for censored instances, which can lead to inaccurate predictions for non-censored instances. This limitation can have important implications for decision making based on the model's predictions, especially if the non-censored instances represent high-value assets.

**Table 1.** Right-censored machines in the dataset.

| Censored Instances | Non-Censored Instances | Proportion of Censored Instances | Proportion of Censored Instances |
|:---:|:---:|:---:|:---:|
| 9661 | 339 | 0.9661 | 0.0339 |

### 4.3. Types of Machine Failure

Identifying the type of machine failure is of utmost importance, as each failure mode can have a varying impact on production efficiency, cost, and uptime. Proper identification can help with the effective allocation of resources, maintenance scheduling, and implementing appropriate mitigation measures. Understanding the underlying causes of different machine failure types and accurately identifying them is crucial for the efficient operation of a machine-dependent system.

### 4.3.1. Tool Wear Failure (TWF)

Tool wear failure is a gradual process that occurs because of continuous use of the machine, thus creating a strain on its components, causing it to fail. This type of failure can impact the quality of results produced by the machine and can result in increased replacement costs. Identifying the factors contributing to tool wear can help to optimize the use of tools and reduce the frequency of replacements.

### 4.3.2. Heat Dissipation Failure (HDF)

Heat dissipation failure occurs when there is insufficient cooling of the machining process, leading to an increase in the temperature of the tool and workpiece. This can cause deformations, cracks, or even melting of the material being machined. HDF is particularly important in high-speed machining operations, where the heat generated can be significant and quickly damage the tool or the workpiece.

### 4.3.3. Power Failure (PWF)

Power failure (PWF) happens when the machining process is not supplied with enough power or when there is an excessive amount of power being delivered. Insufficient power can lead to a reduction in the material removal rate, while excessive power can cause the tool to break or lead to workpiece damage. PWF is an important failure mode to monitor because it can directly impact the productivity and quality of the machining process.

### 4.3.4. Overstrain Failure (OSF)

Overstrain failure occurs when the forces acting on the tool exceed its design limits, leading to deformation or even breakage of the tool. This can happen when the material being machined is particularly hard or when the machining parameters are not optimized. OSF can lead to significant downtime and repair costs, if not prevented.

### 4.3.5. Random Failures (RNF)

Random failures occur due to factors outside of the control of the machining process, such as material defects, operator error, or environmental factors. RNF can be particularly challenging to predict and prevent, making it important to continuously monitor the machining process and identify any patterns or trends that may indicate a potential failure.

## 5. Results

The model was tested for its accuracy performance in the machine failure prediction, while also in the prediction of the type of machine failure. This means that the feature ranking/selection layer was also used to measure the best features to calculate each machine failure mode.

### 5.1. Model Evaluation Criteria

#### 5.1.1. C-Index

Assessing the accuracy of machine failure prediction models can be challenging due to the presence of right-censored failure events in the testing dataset. This means that some machines may have been operational for the entire duration of the testing period, making it impossible to observe when they would have failed. To address this challenge, a common metric used to evaluate the performance of machine failure prediction models is the concordance index (C-index).

The C-index considers both the observed failure times and the predicted failure times, including those that are censored. This is achieved by creating pairs of machines, where the machine with an observed failure time is ranked higher than the machine with a censored failure time. The C-index ranges from 0.5 to 1, with a value of 1 indicating perfect prediction performance, where the observed failure times follow the same order as the predicted failure times, and a value of 0.5, indicating that the prediction model performs no better than random chance. Therefore, the C-index provides a reliable way

to assess the accuracy of machine failure prediction models, even in the presence of censored data [22].

Calculate the total C-index by summing all values and dividing by the total number of possible pairs:

$$C = \frac{1}{num} \sum_{i:d_i=1} \sum_{j:y_i<y_j} I[\hat{y}_i > \hat{y}_j] \tag{10}$$

where $C$ represents the C-index; $num$ = number of all comparable pairs; $y_i$ and $y_j$ represent the observed and predicted time to fail, respectively; and $I$ is the indicator function. Therefore, this metric includes censored data, creating ranked pairs where the observed, uncensored events occur before the observed censored event.

### 5.1.2. Percentage Change

Percentage change is a mathematical calculation that shows the difference between two values as a percentage of the original value. To calculate the percentage increase, the difference between the new and old values must be found and divided by the original value. This difference is then multiplied by 100 to obtain the percentage increase. The formula used to make this calculation is presented in Equation (12).

$$I = y - x \tag{11}$$

$$Percentage\ Increase = \frac{I}{x} \cdot 100 \tag{12}$$

In Equation (11), $x$ is the number before the increase, $y$ is the number after the increase, and $I$ is the increase between the two values. In Equation (12), $I$ is the increase between the values and $x$ is the original number.

### 5.2. COX and RSF Feature Analysis

For the selection of the most important features, a set of feature selection and feature analysis techniques were used. These results were then compared and the features that prevailed were applied to the RSF model for training. The features were also ranked for each of the machine failure types mentioned in Section 4.3.

The COX and RSF methods in the feature ranking section showed that the most important features in terms of machine failure were torque, rpm, and air temperature, as shown in Figure 2 and Table 2. Specifically, in the figure and table, the ranking of the characteristics is shown for the type of machine failure. In the COX results, the values that have a negative value indicate low correlation with machine failure, while positive values indicate an increased probability of machine failure. In contrast, in the RSF results, negative values indicate that the characteristic reduces the predictive ability of the model, i.e., in this specific case, the correlation of the characteristic with a low risk of machine failure. In the case of RNF prediction, all characteristics have negative values, indicating that they are all related to a low risk of machine failure. Although this complicates the model prediction, including the context of machine failure, i.e., a random factor, it is somewhat logical.

**Table 2.** Feature ranking results based on RSF importance mean.

| Feature | Machine Failure | HDF | OSF | PWF | RNF | TWF |
|---|---|---|---|---|---|---|
| Rotational speed [rpm] | 0.1173 | 0.0850 | 0.1594 | 0.0047 | −0.1413 | 0.1377 |
| Torque | 0.0678 | −0.0008 | 0.0028 | 0.2343 | −0.1162 | 0.1041 |
| Air temperature [K] | 0.0480 | 0.0731 | (~)−0.0000 | (~)−0.0000 | −0.0623 | −0.0194 |
| Process temperature [K] | 0.0070 | −0.0098 | 0.013 | 0.0015 | −0.1022 | 0.0161 |
| Type | 0.0006 | −0.0001 | 0.0062 | (~)−0.0000 | −0.0558 | −0.0026 |

The ranking of the features in the various techniques differed slightly, but the characteristics were consistently among the top-ranked features. Specifically, the features of torque and rpm were consistently ranked among the top three features.

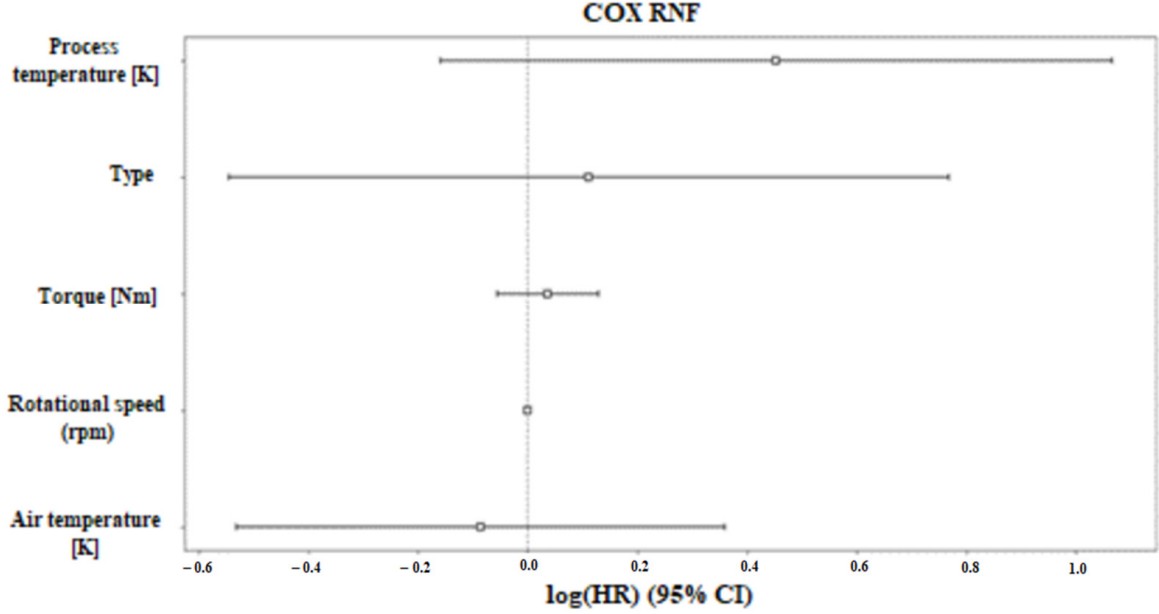

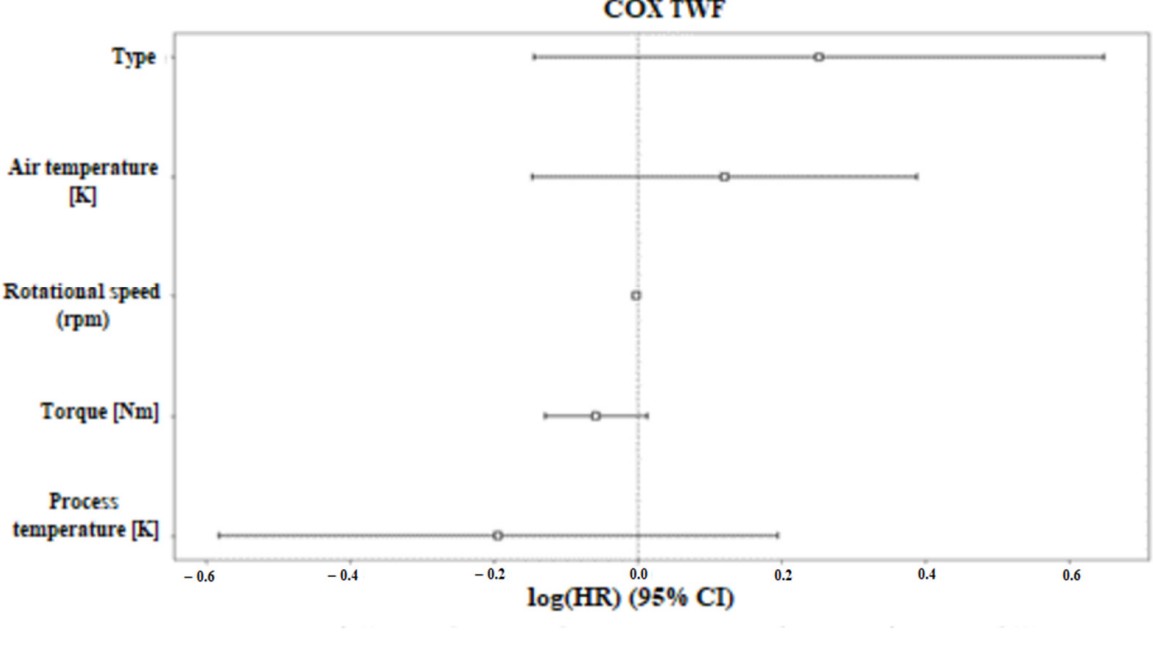

(**a**)

**Figure 2.** *Cont.*

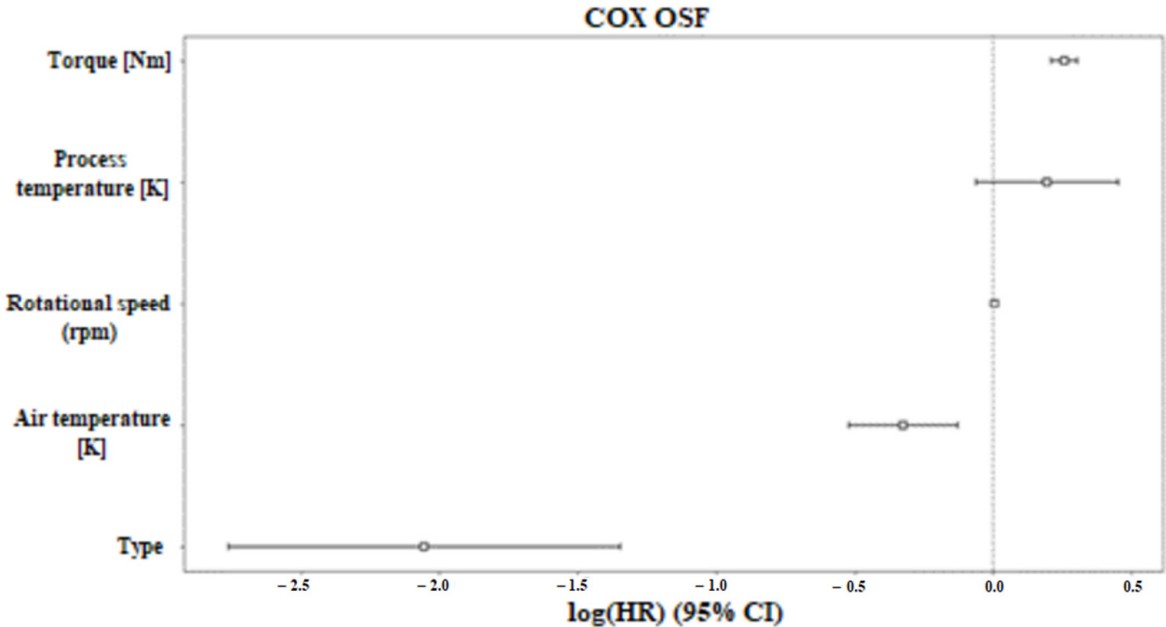

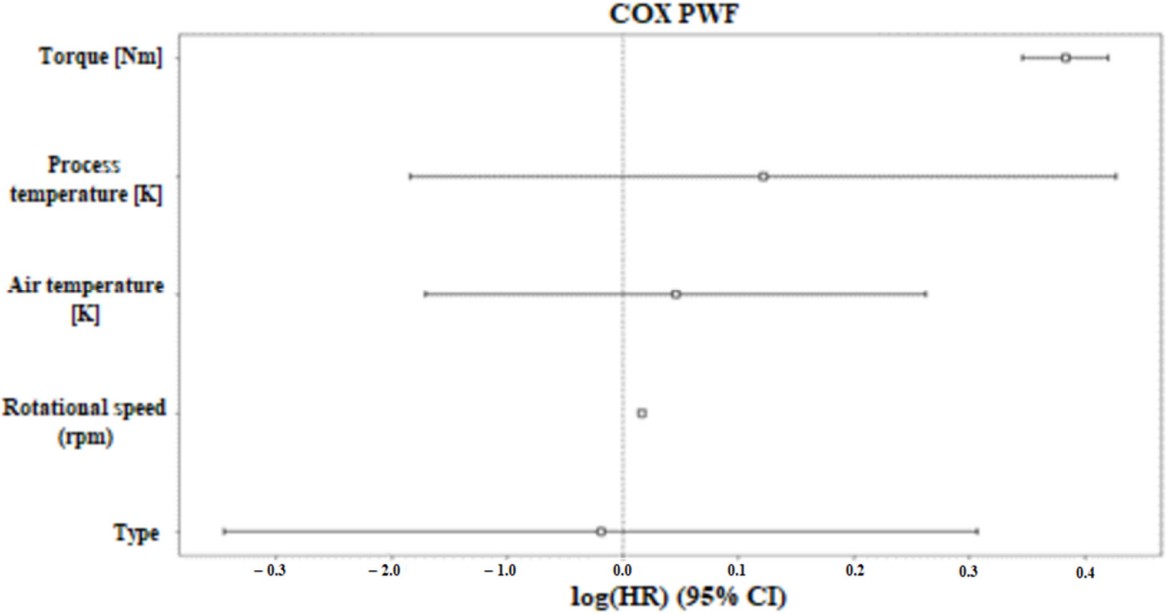

**(b)**

**Figure 2.** *Cont.*

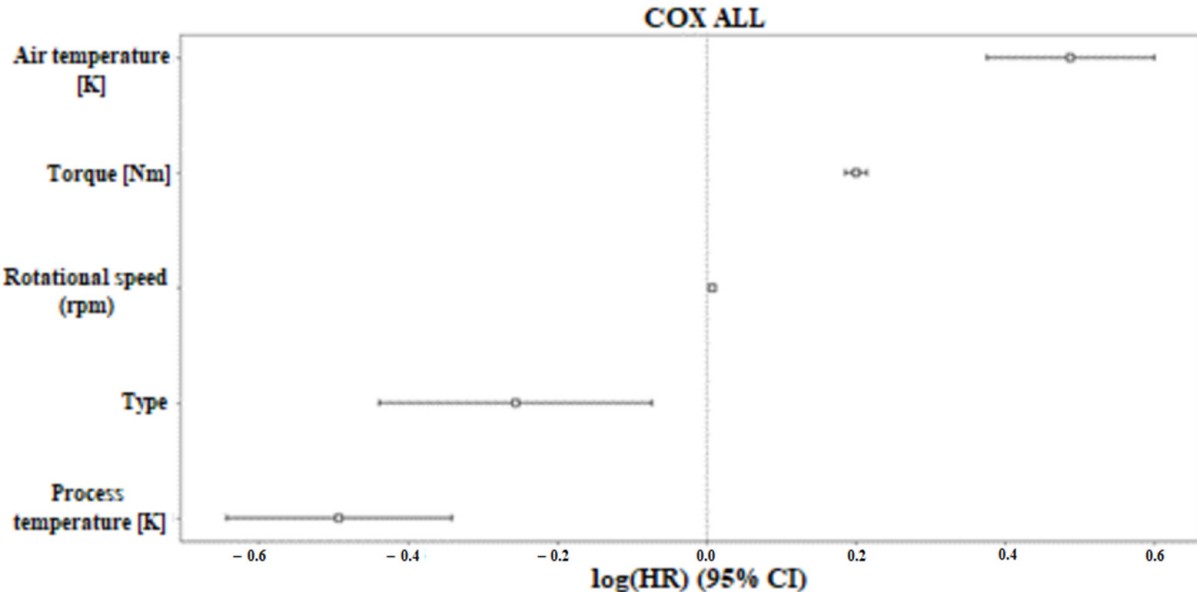

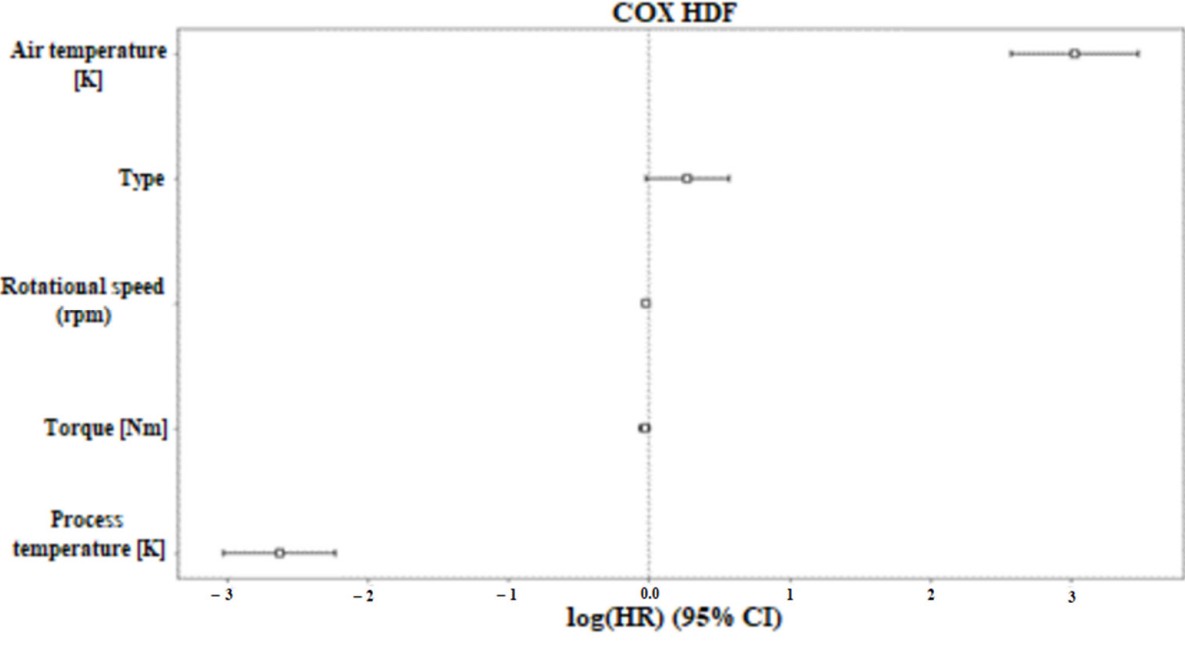

**(c)**

**Figure 2.** (**a**) Cox feature analysis results for RNF and TWF; (**b**) OSF and PWF; (**c**) and for machine failure and HDF.

For the COX results shown in Figure 2, we can observe that in the case of all the failed machines, the COX diagram showed a strong correlation between the machine failure risk

and the increase in the air temperature. Moreover, the torque feature seems to have a correlation with the failure of the machine but to a smaller degree. The rest of the features in the diagram seem to have either no impact (rotational speed) or are associated with a lowered risk of the failure of the machine. Similar results are shown in the HDF failure type on the COX diagram where the air temperature seems to have the highest correlation to the risk of machine failure, while the increase values in other features seem to have a lower risk. For the OSF, RNF, PWF, and TWF failure types, all the features seem to have a small correlation with the event we aimed to capture, so using the other metrics in combination with the COX diagram is needed to better evaluate the features. The small correlation is noticed because of the small values of the positive and negative rankings, which means that most of the features are closer to 0.

### 5.3. RadViz and Rank2D Depiction of Feature Relations

For a better understanding of the features, feature analysis techniques such as the RadViz and Rank2D technique were used. Through them, the relationships of the characteristics between them can be analyzed and better understood, which provides better results in their sorting.

In case of machine failure, the RadViz analysis, depicted in Figure 3, showed that a high rotational speed combined with a high process temperature is a possible factor contributing to the failure. This is evident as only failed machines are pulled towards the rpm feature, with some being close to the middle of rpm and the process temperature. The same results can be observed in the PWF case, where most of the failed machines in the machine failure category that were pulled towards the rpm feature seem to belong in PWF. So, we can assume that increased rpm is correlated with a PWF type of machine failure. Apart from that, we can see that most of the machines, working or otherwise, are closer to the torque feature and the area between the torque and the type of machine. The strong pull towards the torque feature indicates that torque is critical for the proper functioning of the machines. The fact that many of the machines are also located in the area between torque and type suggests that there may be a relationship between the type of machine and its torque output. Finally, the light pull towards the process temperature for both working and failed machines indicates the importance of this feature, especially in the failure categories of PWF, TWF, and RNF, where some of the failed machines can be seen moving towards that area of the diagram.

Through the analysis of 2D ranking, a strong correlation between the process and air temperature features with the rpm of the machine is observed, which is logical in the mechanical context. Additionally, increasing the rpm of the machine results in a corresponding decrease in its torque, as shown in Figure 4. Moreover, using the Pearson ranking 2D model, as shown in Figure 5, we observe an almost perfect positive correlation between the air temperature and process temperature features. This leads us to the conclusion that an increase in one of these two variables may lead to the destruction of the machine. Similarly, the almost complete negative correlation between the rotational speed and torque variables is also evident from the same diagram. This negative correlation may indicate that an increase in one of the two variables results in a decrease in the probability of machine failure. These observations highlight the importance of considering these variables in machine failure prediction models.

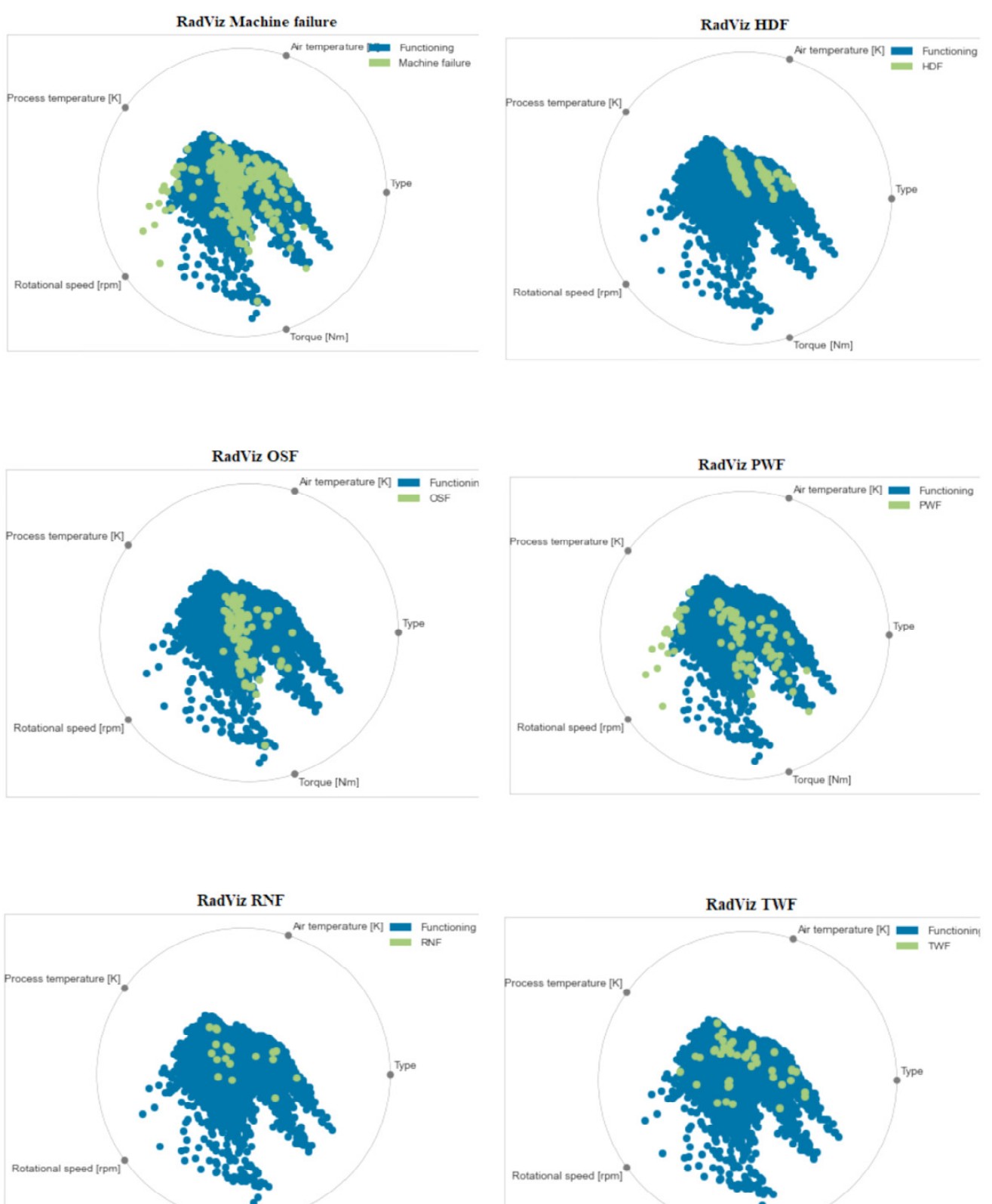

**Figure 3.** RadViz graphical representation of feature relations for different causes of machine failure.

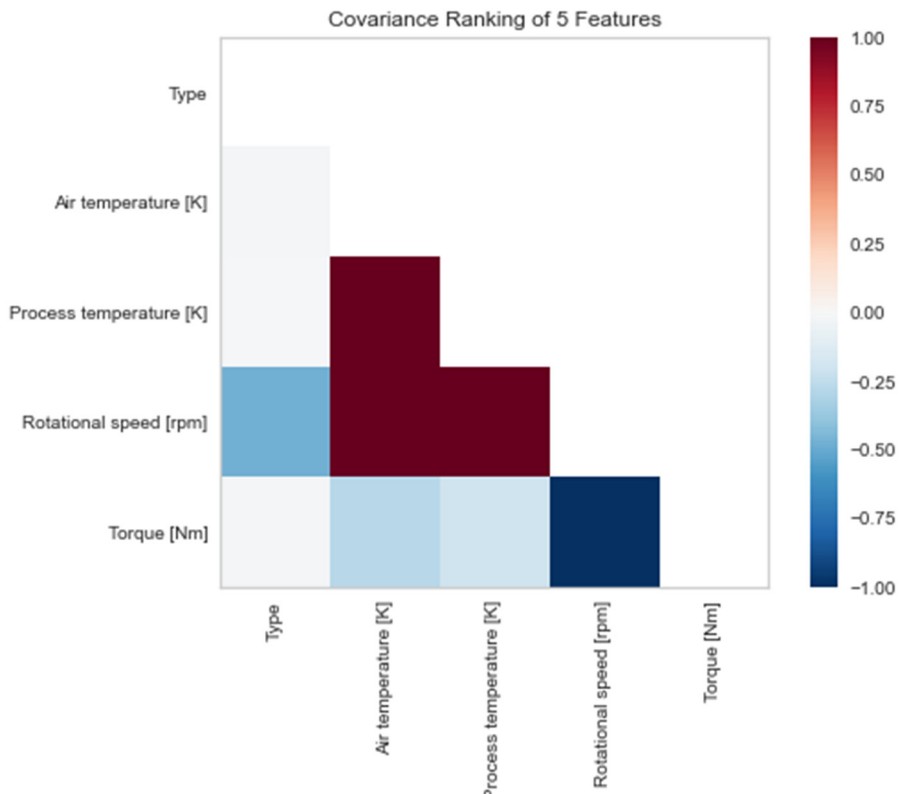

**Figure 4.** Results of linear correlation of features using covariance 2D ranking.

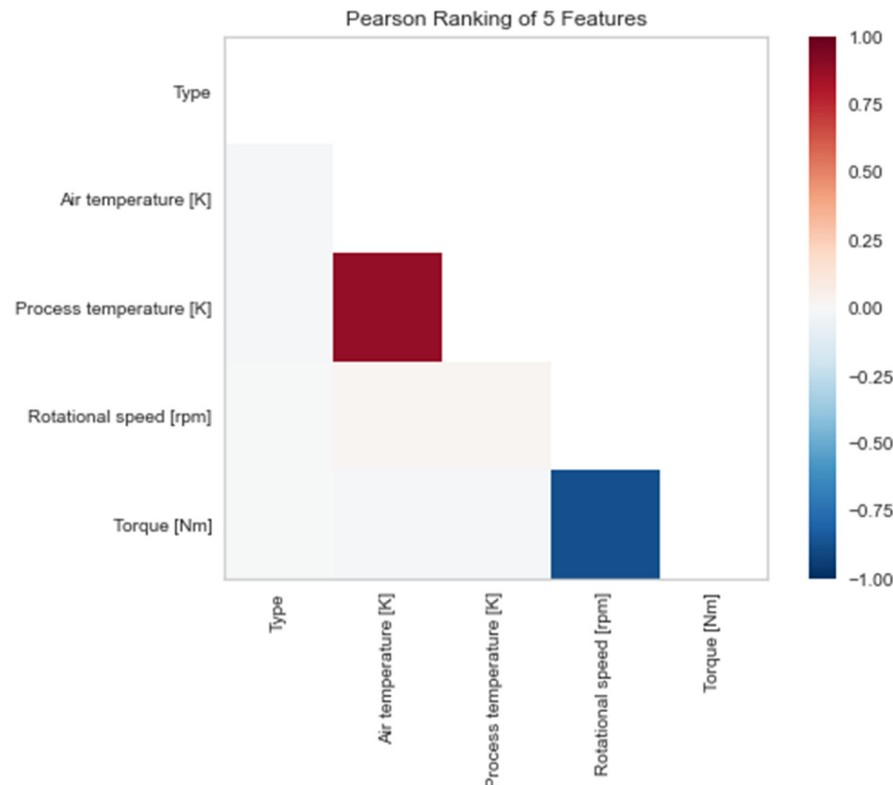

**Figure 5.** Results of linear correlation of traits using Pearson 2D ranking.

### 5.4. Number of Selected Features

The recursive feature elimination technique was used to select the number of characteristics to be used in the final prediction model. Based on this, as well as the results of the negative influence of features obtained from RSF and COX, we can formulate the characteristics that will give us the best possible prediction results.

According to the results for the general prediction of machine failure, RFE used two features to achieve the best prediction, as shown in Figure 6. The addition of more features reduced the prediction accuracy, which is normal considering the previous feature ranking evaluation. However, if we also consider the results of the RSF and COX, we can be more generous with the number of features that will be used, by adjusting their quantity to a number quite close to the original number given by RFE. The same methodology can be used for predicting the type of machine failure.

Furthermore, in Figure 6, it was observed that in most cases, the best prediction accuracy was achieved with only two features. The only exceptions were the TWF case, which required three features, and the OSF case, which required all five features to achieve the best possible prediction. The decrease in the precision with the addition of more features could be an indicator that the most relevant information for the prediction was contained within a small subset of the available features.

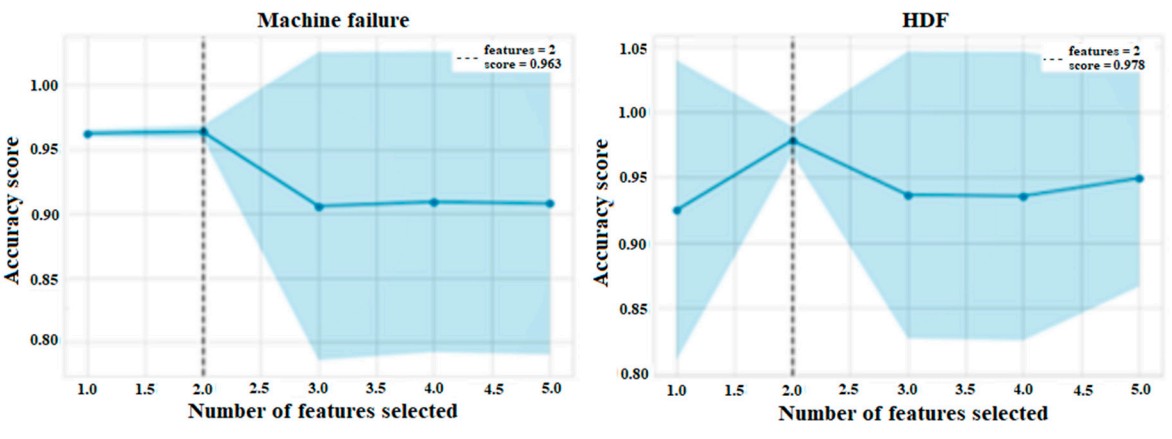

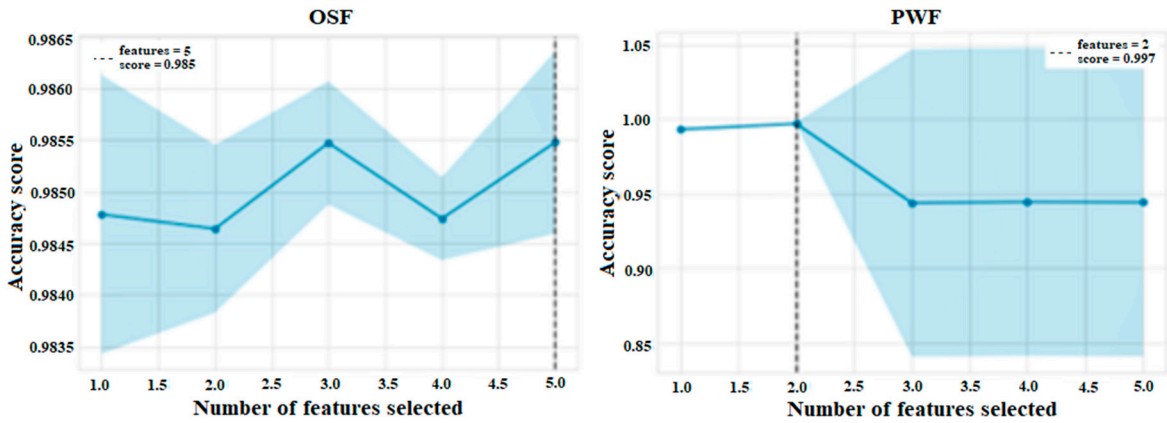

**Figure 6.** *Cont.*

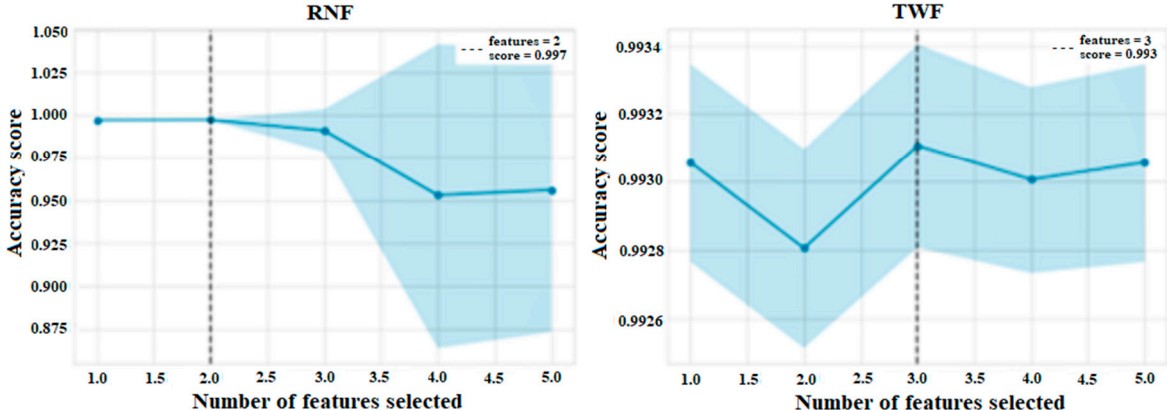

**Figure 6.** Optimal number of features based on RFE accuracy score for different causes of machine failure.

### 5.5. Survival and Cumulative Hazard Curves

From the analysis of the survival and cumulative hazard curves, it is observed that the safe operation of the machines ranges from 0 to ~200 min of operation. From this point onwards, the risk of machine failure increases rapidly. The same pattern is confirmed by the failure of machines after this time point, as shown in Figure 7. Furthermore, an examination was carried out on the machines in the only categorical variable contained in the dataset, which was the type of machine shown in Figure 8. This variable took on the values of high, medium, and low. From this examination, it appears that high-quality machines perform better than low- and medium-quality machines, although to a small extent, in the time range of 200–240 min. This may also be due to their relatively smaller population, as high-quality machines accounted for 20% of the total dataset. In addition, a noticeable superiority of medium-quality machines over low-quality machines is observed in the time range of 200–230 min, which evens out in the later time frames.

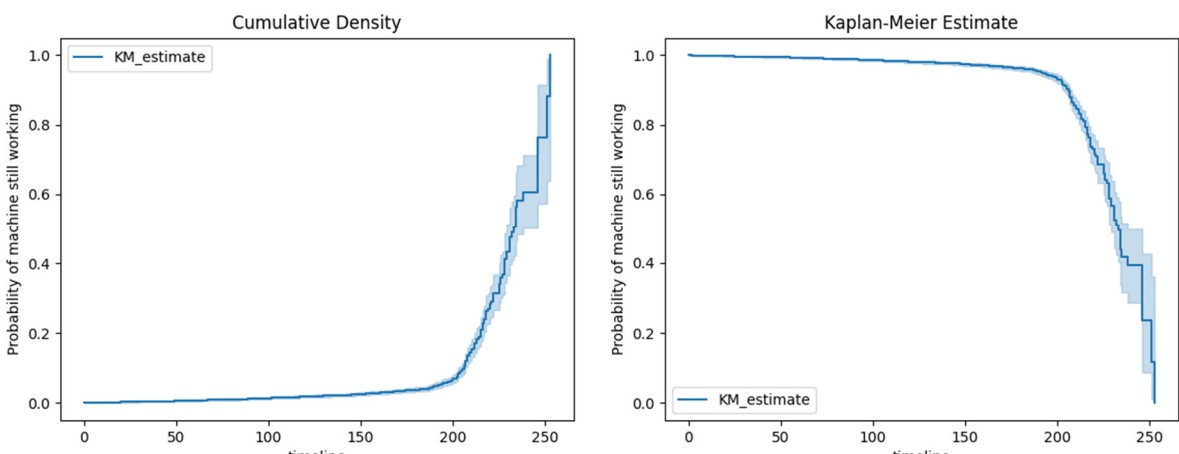

**Figure 7.** Cumulative hazard and survival curves.

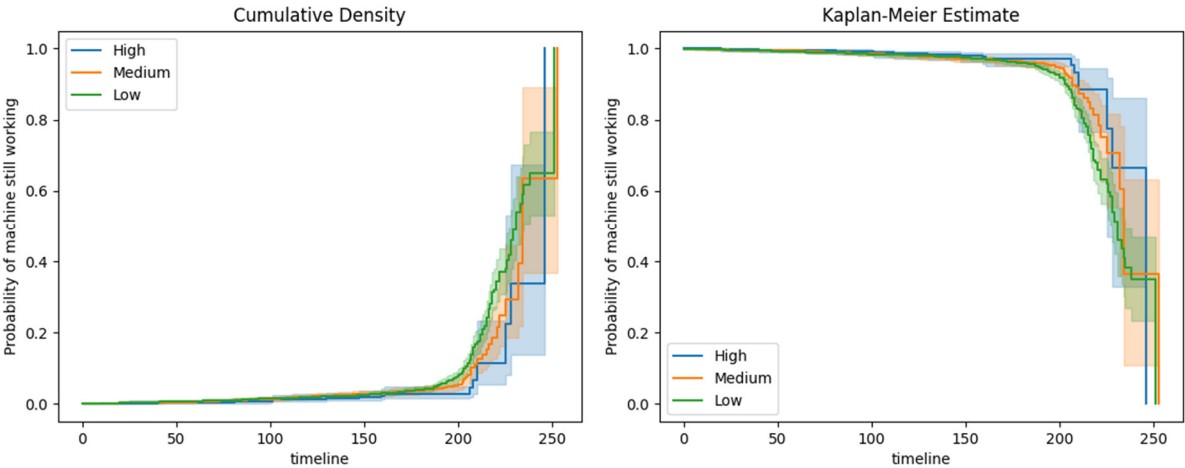

**Figure 8.** Cumulative hazard and survival curves based on machine type.

*5.6. Model Training and Results*

For our evaluation, we measured the predictions of the RSF model against a normal random forest model, which was trained with the exact same dataset and configuration as the RSF but with the downside of ignoring the right-censored data, unlike in RSF. We used the models one time to make the prediction using the whole dataset (Table 3) and its features and another time with the feature selecting layer (Table 4). In both, we used the metric of the C-index to calculate the accuracy of each model. It is important to note that both models were trained using identical hyperparameters for the calculation of the C-index.

**Table 3.** Model results for machine failure and all its causes with the whole dataset, using C-index.

| Model | Machine Failure | HDF | OSF | PWF | RNF | TWF |
|---|---|---|---|---|---|---|
| RSF | 0.9724 | 0.9976 | 0.9735 | 0.9936 | 0.6695 | 0.5446 |
| RF | 0.9558 | 0.9145 | 0.9956 | 0.9986 | 0.3899 | 0.9222 |
| COX PH | 0.8837 | 0.9944 | 0.9957 | 0.9965 | 0.3931 | 0.9642 |
| ST | 0.8447 | 0.9399 | 0.8868 | 0.9475 | 0.4972 | 0.6302 |

**Table 4.** Model results for machine failure and all its causes with feature selection, using C-index.

| Model | Machine Failure | HDF | OSF | PWF | RNF | TWF |
|---|---|---|---|---|---|---|
| RSF | 0.9724 | 0.9977 | 0.9735 | 0.9951 | 0.7644 | 0.6841 |
| RF | 0.9627 | 0.9093 | 0.9958 | 0.9987 | 0.4792 | 0.9217 |
| COX PH | 0.8983 | 0.9672 | 0.9955 | 0.9965 | 0.8603 | 0.9718 |
| ST | 0.8476 | 0.6457 | 0.9420 | 0.9670 | 0.4967 | 0.6874 |

The data were divided into a 75% training and 25% testing set to evaluate the performance of our model. For the RSF, we selected the splitting criterion based on the log-rank test, which compares survival curves between two or more groups. The data were applied to the RSF model consisting of 1000 trees. To make a prediction, a sample descends tree by tree down to a terminal node in the forest. The data in each terminal node are used for non-parametric estimation of the survival function and cumulative hazard using the Kaplan–Meier and Nelson–Aalen estimators, respectively. Additionally, the risk score representing the expected number of events for a particular terminal node can also be computed. The overall prediction is the average of all the predictions of the trees in the forest. The model produced very accurate prediction results for machine failure, about 97%, as well as when the machine failed due to heat dissipation failure (HDF~99%), overload

(OSF~97%), and lack of power supply (PWF~99%). In cases of random failure (RNF) and tool wear failure (TWF), the model did not predict the outcome as efficiently, achieving a C-index of ~54% for RNF and ~0.67% for TWF. This fact is likely due to the lack of sufficient machines that have stopped working due to these causes, as well as the more difficult nature of predicting them.

The results show that the RSF model has the upper hand in the main prediction of the machine failure out of all the models tested but lacks in the prediction of some types of failure. In the failure prediction and without using the data analysis/selection layer, the model that kept up with the RSF in terms of C-index score was random forest with a score of 0.9558, only 0.0166 less than the C-index of RSF. Both COX PH and ST scored below 90%. As for the types of failure, the results showed an imbalance among the predictions of the models, where no model excelled in the prediction of all the types. For the HDF and RNF failure types, the best prediction was achieved by the RSF with a percentage of approximately 99% and 67%, respectively, while for the failure types of OSF and TWF, the COX PH model displayed the best C-index score among all the other models, achieving scores of 99.57% and 96%, respectively. Finally, the RF model produced the best C-index score for the PWF failure type with 99.86%. The numbers discussed are presented in Table 3.

With the insertion of the feature selection layer, all models increased their predictions in most cases. It is noteworthy that the optimal set of selected features used by the models differed across the various modes of failure. The features used by each model to achieve its best C-index score can be seen in Table 5. All models used on average four features, with the decimal values rounded up to the closest integer because the features cannot be split. One outlier to this was the prediction of TWF, where three out of the four models used two features to achieve their best prediction, while the RF model used all five.

**Table 5.** Number of features each model used to achieve its best prediction on each failure type.

| Model | Machine Failure | HDF | OSF | PWF | RNF | TWF |
|---|---|---|---|---|---|---|
| RSF | 5 | 4 | 5 | 3 | 3 | 2 |
| RF | 3 | 4 | 5 | 2 | 2 | 5 |
| COX PH | 3 | 4 | 5 | 5 | 3 | 2 |
| ST | 5 | 4 | 2 | 5 | 2 | 2 |

The RSF model showed significant improvements in the prediction of TWF (from ~54% to ~68%) and RNF (from ~67% to ~76%) types, while also making slightly better predictions on the PWF, OSF, and HDF types. The RF model also improved its results both in the machine failure prediction (increase by ~1.1%) and in the prediction of the failure types of RNF, PWF, and OSF. In the case of RNF, the increase was significant as the RF model was almost 10% more accurate using the feature selection. In contrast, the prediction of the TWF and HDF showed a small decrease in accuracy. The COX PH model exhibited improvements in most performance categories, with the exception of PWF, where its performance remained stable, and HDF, where the C-index decreased by 0.0272. The most significant increase was the RNF prediction where the C-index score improved from 39% to 86%, showing a high 119.9% increase. Finally, ST also made major improvements in the prediction of the machine failure, as well as the OSF, PWF, and TWF failure types. It was also the model with the most significant decrease in the C-index score after the feature analysis/selection layer, with the score of HDF decreasing by approximately 31%. Another drop-off was also present in the RNF failure type, although this was less significant. The percentages of increase and decrease in the C-index scores of each model are presented in Table 6.

Considering all models, the RSF, despite feature selection, did not manage to surpass the results of the RF in the predictions of OSF and PWF, and in the predictions of the COX PH model on the RNF and TWF failure types. Especially in the case of TWF, the difference

remained significant despite the improved accuracy. However, RSF performed better in the overall prediction of failure but also stayed close or performed better than the other models in all cases except TWF.

**Table 6.** Percentage of increase/decrease in C-index score after the feature analysis/selection layer.

| Model | Machine Failure | HDF | OSF | PWF | RNF | TWF |
|---|---|---|---|---|---|---|
| RSF | 2.609% | 0.01% | 0% | 1.508% | 14.152% | 25.656% |
| RF | 0.757% | −0.455% | 0.02% | 0.1% | 22.937% | −0.054% |
| COX PH | 1.744% | −2.732% | −0.02% | 0% | 119.9% | 0.771% |
| ST | 0.344% | −31.277% | 6.043% | 2.174% | −0.1% | 8.951% |

## 6. Discussion

The results show that the RSF model performs better than the other models selected for this study in predicting the main cause of machine failure, achieving a C-index of 0.9724. In general, the feature selection layer had a mostly positive impact on the C-index score in all the models, as depicted in Table 6. The table was calculated using the method described in Section 5.1.2. The most impressive increase was seen in the COX PH model, which more than doubled its C-index score. Furthermore, the model that seemed to be favored the most by the addition of the feature selection layer was RSF, which is the only one that either increased or kept the same C-index score. In contrast, the addition of the feature selection layer exhibited a negative impact on the prediction of the HDF failure type in three out of the four models that were tested. The most significant decrease of this failure type was noted in the ST model, whose C-index score decreased by approximately 31%. This shows that further research may be needed to address this issue in this specific type of failure, as it seems to be more susceptible to changes in the data.

One notable finding from our study is that most models demonstrated improved accuracy in predicting failure types that were previously challenging to evaluate, suggesting a substantial enhancement in their predictive capabilities. This improvement is attributed to the incorporation of a feature analysis and selection layer, which effectively identified and leveraged the most relevant features for predicting these challenging failure types. This was mainly in the RNF and TWF failure types, where the models demonstrated lower C-index scores. For RNF, all models showed increases of over 14%—apart from the ST model, which had a slight decrease of 0.1%. Regarding TWF, the models that had the worst performance, namely the RSF and ST, had an increase of 25.65% and 8.9%, respectively. These increases in C-index score in the models show that the feature selection layer critically improves the overall prediction, helping it reach a more consistent prediction accuracy among all the failure types. It must be noted that the limited number of features provided in the dataset used may be a restricting element of the overall increase that the feature selection layer can provide to the model. Moreover, as emphasized in the Results section, the decrease in the C-index score of the HDF failure type is of major importance, and further research is needed to address this issue.

One limitation of this study is the exclusive use of a single dataset. While the dataset was selected based on its suitability for addressing the research questions and objectives, the findings may not be generalizable to other populations or contexts. Additionally, the dataset contained a high number of right-censored data, with 96.61% (Table 1) of the instances being censored. This could impact the accuracy and reliability of the results, as well as limit the methods that were used. It is important for future research to consider using multiple datasets and exploring alternative methods for addressing censored data to improve the validity and generalizability of the findings.

## 7. Conclusions

The breakdown of machines is a major issue in today's technology-dependent societies. To better manage this problem, we propose a machine failure prediction model based on

survival analysis techniques. The developed model filters the main features that contribute to machine failure through RSF, COX regression, Rank2D, and RFE techniques. Then, the selected data are fed into the RSF model, which is used to perform the prediction. Using RSF in this scenario provides better management of censored data, as machines failure datasets are often heavily right-censored.

The model was subjected to a comparative analysis against a conventional random forest model, survival trees model, and COX PH survival analysis model. It exhibited superior performance in two out of the six categories, while consistently producing reliable predictions, and demonstrating comparable C-index scores with the top-performing models, on the rest of the categories. Additionally, the use of this model enables support for high-dimensional data, which is a common occurrence in machine breakdown. The model showed promising prediction results with a success rate of approximately 97%, while also demonstrating a high ability to predict the cause of the failure of the machine.

The high level of accuracy achieved by the proposed approach makes it an especially valuable tool for the development of fault-tolerant systems in large-scale environments, including those within the Internet of Things (IoT). The ability to accurately predict machine failure can help to prevent costly downtime and minimize the risk of catastrophic system failures, thus enhancing the overall reliability and stability of these complex systems. The proposed approach provides a promising solution for the performance optimization and security enhancement of large-scale systems, with potential benefits spanning a wide range of industries and applications.

While our study provides valuable insights into the performance of the predictive models in the specific dataset used, there are limitations that warrant further research. Future work could involve the evaluation of our proposed approach on a variety of different datasets to assess its generalizability and robustness. Additionally, further optimization of the feature analysis and selection layer could be considered. These efforts could lead to the development of an even more accurate and reliable predictive model.

**Author Contributions:** Conceptualization, D.P., K.D. and N.T.; methodology, D.P., K.D. and N.T.; software, D.P.; validation, D.P.; formal analysis, D.P., K.D. and N.T.; investigation, D.P.; data curation, D.P. and K.D.; writing—original draft preparation, D.P., K.D. and N.T.; writing—review and editing, D.P. and K.D.; visualization, D.P., K.D. and N.T.; supervision, K.D.; project administration, K.D. All authors have read and agreed to the published version of the manuscript.

**Funding:** This research received no external funding.

**Institutional Review Board Statement:** Not applicable.

**Informed Consent Statement:** Not applicable.

**Data Availability Statement:** Dataset supporting reported results can be found in https://archive.ics.uci.edu/ml/datasets/AI4I+2020+Predictive+Maintenance+Dataset (accessed on 8 November 2022). The code supporting the findings of this paper is stored in https://github.com/padimitrios/Machine_Failure_Analysis.

**Conflicts of Interest:** The authors declare no conflict of interest.

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
