# Peer review of "Machine Failure Prediction Using Survival Analysis"

_futureinternet, doi:10.3390/fi15050153_

Round 1

Reviewer 1 Report

The paper explores the application of Survival Analysis and Machine Learning techniques for predicting the risk of machine failure over time. The approach described in the paper addresses an important challenge in industrial settings where machine downtime can result in significant financial losses. The use of Survival Analysis is particularly notable as it enables the prediction of the time-to-event, which in this case is the failure of a machine. Unfortunately, the method has limited scientific impact on the area, as the main proposal is the application of feature selection and analysis techniques in conjunction with a Random Survival Forest (RSF) model. Furthermore, the addition of preprocessing steps to the RSF only shows a significant advantage over a regular Random Forest in a very strict set of failure types.

I would also like to recommend some points to improve the paper’s structure and readability:

1) In line 14, there is a bit of confusion (which reverberates throughout the paper) about the proposal. It needs to be clearer if the work is proposing a statistical analysis, a model, an approach, or a combination of them to “predict” machine failure. On line 22, the phrase “In this work, a statistical data analysis was carried out, with the aim of predicting machine failure using the Survival Analysis,” can give the impression that your proposal is simply an analysis of this dataset to find patterns to predict machine failure, which is not the case. The paper proposes an approach with feature analysis and a survival analysis model, among other steps, that is analyzed and discussed at the end.

2) Revise writing errors, such as the loose phrase “they shield” on line 12 and Random Survival in line 21.

3) In the Introduction of the paper, it is important to better contextualize the reason for choosing Survival Analysis and Random Survival Forest with more references, especially if you are going to compare them to other kinds of approaches that are present in the literature.

4) The contributions list could be placed after the motivations.

5) The paper has a very unusual structure, where Related Works, Proposed Method, and the following applied techniques are all in the same section of Materials and Methods. I would suggest that Related Work should be its separate section. The paper would benefit as well from a Theoretical Background, where all the theoretical information currently dispersed throughout the paper could be condensed.

6) Add a paragraph in Related works comparing the proposed approach to the ones found in the literature, highlighting the possible advantages and applications. This strengthens your proposal and helps to better assess its impact.

7) There is an underlying issue with the paper that is talking about certain things before properly introducing them. For instance, in the Related Works, you start the section by citing the usage of “Kaplan-Meier curves and the Cox Regression” in the literature without explaining what they are or why you are talking about them since you only explain that your approach uses them later on the paper. If you discuss them early on, it can help to give context not only of the problem but also on how you propose to solve it.

8) The Proposal Model section (2.2) mainly focuses on the motivation/contribution of your paper and barely discusses the proposal itself. Instead of presenting an item-list of the overall steps of your approach, I recommend adding at least a paragraph for each step to better describe and associate them with the presented diagram.

9) The Approach Workflow Diagram needs to be reevaluated as there seems to be a mismatch between the described approach in the following sections and the steps presented on the diagram. Some techniques are not shown, for instance, the techniques of the 2.3 Section, while others are repeated on the diagram.

10) In line 301, Feature Selection is described as an important step in classification solutions, but your approach is a Survival Analysis model. It would be clearer to describe the feature selection as an important step in the RSF model, rather than in classification solutions.

11) Move the Database section to the Materials and Methods section, as it is more appropriate to be included there.

12) It would be helpful to discuss the types of failures before presenting their results, as it would give the reader a better context for interpreting the results.

13) Establish a ranking criteria for the features, such as "we select the top-3 features for each technique" to help readers understand how the features were selected.

14) Figure 2's labels are almost unreadable. Please consider increasing the font size or using clearer labels to improve readability.

15) It would be interesting to analyze the size proportion between censored instances and non-censored instances in the database section, as this information could help to explain any biases or limitations in the dataset.

16) In line 395, it is written "cannot" where I think you meant "can". Please double-check for similar errors throughout the paper.

17) Figure 6's score label should indicate which score they are referring to, such as "Accuracy Score" or "AUC Score", to avoid confusion.

Reviewer 2 Report

Summary:

The author used Feature Selection and Random Survival Forest methods to predict machine failure. Based on the experiments, the author claimed this approach is better than the Random Forest method.

I have some comments below:

-       The title is too general, which the proposed method claimed to solve common machine failure prediction. However, the manuscript only used one (limited) dataset to assess the proposed method’s performance.

-       In Figure 1, there are several redundant components in each layer. Thus, it makes the proposed models inefficient enough.

-       The proposed method seems complex and involves several redundant components, while its explanation is hard to follow. Thus the reader can fail to capture its essence. Please consider a top-down style to describe/discuss the proposed method.

-       I am concerned about the data used in the experiment, where in the real world, the machine failure data is commonly categorized as imbalanced data, where the number of failed machines is just a minority.

-       In tables 2 and 3, do RSF and RF have the same preprocessing technique and features? And why didn’t the authors compare it with the deep learning-based models, which are recently well-known as a state of art prediction task?

-       Section 4 does not discuss the experimental result. It just summarizes the manuscript.

-       The authors should provide a conclusion summarizing their findings, the method’s advantages, and limitations. Please add a conclusion section.

Reviewer 3 Report

In this paper, a statistical data analysis was carried out, with the aim of predicting machine failure. A new approach based on the Random Survival Forest model and an architecture that aims to filter the features that are of major importance to the cause of the machine failure, is proposed to more accurately and consistently predict the mentioned event. Generally, the quality of the English need to be improved. Many errors. The novelty should be highlighted. It is vague and I can’t see any novelty in the paper. The literature review section that discusses related works should discuss limitations of previous methods and their numerical findings and should be well structured.  Also, a comparative study is not performed which is an important step to show the effectiveness of the proposed method.

The abstract The novelty of the paper is vague since survival random forest and filtering algorithms are not novel techniques. Could you please explain more on your proposed method in the abstract since as you mentioned this is your contribution and as you know that are previous papers the used random survival forest and filtering approaches for survival analysis. Please revise the English.

What does “they shield” mean in line 12, I think it is a typo?

Introduction:

In the introduction section you mentioned in line 78, that in the proposed model, state-of-the-art survival analysis techniques are used. Random survival forest and filtering are not state of the art techniques. Survival analysis methods based on deep learning such as recurrent networks like LSTM, Bi-LSTM, and CNN-LSTM are state of the art.

You have to mention what motivated you to use Random survival forest and filtering algorithms like the advantage of each of them. Discuss the privilege of using these algorithms techniques over other survival analysis techniques.

This section should end with a paragraph describing paper organization.

Method and Materials  

The related work section should be moved to the introduction section or a separate independent section before the methods and materials section. It should be restructure and rewritten to include limitations of previous studies and the numerical findings

The first paragraph of the proposed model section is unnecessary , authors are advised to remove this paragraph.

The workflow in figure 1 is not explained. Please explain the briefly the steps of the proposed model.

Also, please explain the details of the steps of the proposed model.

There is no novelty in this paper. I can see that you used some of the known methods of survival analysis that were previously used in the literature.

Why did not you used deep learning since they are the state of the art?

Experimental Results

Dataset description should be moved to methods and materials section.

The quality of the figures in the results section is poor. The resolution of the figures needs to be enhanced.

Please add a performance metrics section and define the performance metrics you are using.

There is no clear discussion on the results of the proposed method. This section should be rewritten. Figures and table should be inserted just after their description and discussion of their results. They also should be cited in the text.

Why didn't you compare your results with some of the related studies based on the same datasets you used? This is an important step in any research article to verify the competence of the proposed method.

Also, why didn’t you compare the results of the proposed model with other popular survival analysis methods?

Please add a conclusion section

Please mention the limitations of your technique and your future work.

Round 2

Reviewer 1 Report

The authors presented an improved paper version. The contribution could be improved by selecting fairer baselines and emphasizing state-of-the-art, but I believe the current quality is ok.

Reviewer 2 Report

Thank you for the revision.
Mostly my points on the experiments part were not addressed well, and I could not find a significant improvement after revision.
Thus I could not recommend to accept this paper in present form.

Reviewer 3 Report

First, I would like to thank the authors for improving the quality of the manuscript. However, some comments need to be properly addressed.
I asked the authors the following " The related work section should be moved to the introduction section or a separate independent section before the methods and materials section. It should be restructure and rewritten to include limitations of previous studies and the numerical findings. Please move it just after the introduction section. Moreover, the related work section discussed few studies, more studies should be added to the related work section.

Regarding my comment: Also, why didn’t you compare the results of the proposed model with other popular survival analysis methods? You did not address this comment properly, you can just compare your results based on popular survival analysis approaches such as Cox PH model, MTLSA, linear MTLSA and MTL-ANN, survival trees.

please add a reference citing the dataset.

Since the dataset is publically available, therefore, there should be definitly some studies that employed this dataset for survival analysis. please search for them and compare your results with them

Round 3

Reviewer 2 Report

Thanks for adding the experiments. I recommend accepting this manuscript in its present form.

Reviewer 3 Report

The authors have addressed my comments.

Thank you!